# MAKE-A-VIDEO: TEXT-TO-VIDEO GENERATION WITHOUT TEXT-VIDEO DATA

Uriel Singer [+]      Adam Polyak [+]      Thomas Hayes [+]      Xi Yin [+]

Jie An    Songyang Zhang    Qiyuan Hu    Harry Yang    Oron Ashual    Oran Gafni

Devi Parikh [+]      Sonal Gupta [+]      Yaniv Taigman [+]

Meta AI

## ABSTRACT

We propose Make-A-Video – an approach for directly translating the tremendous recent progress in Text-to-Image (T2I) generation to Text-to-Video (T2V). Our intuition is simple: learn what the world looks like and how it is described from paired text-image data, and learn how the world moves from unsupervised video footage. Make-A-Video has three advantages: (1) it accelerates training of the T2V model (it does not need to learn visual and multimodal representations from scratch), (2) it does not require paired text-video data, and (3) the generated videos inherit the vastness (diversity in aesthetic, fantastical depictions, etc.) of today's image generation models. We design a simple yet effective way to build on T2I models with novel and effective spatial-temporal modules. First, we decompose the full temporal U-Net and attention tensors and approximate them in space and time. Second, we design a spatial temporal pipeline to generate high resolution and frame rate videos with a video decoder, interpolation model and two super resolution models that can enable various applications besides T2V. In all aspects, spatial and temporal resolution, faithfulness to text, and quality, Make-A-Video sets the new state-of-the-art in text-to-video generation, as determined by both qualitative and quantitative measures.

## 1 INTRODUCTION

The Internet has fueled collecting billions of (alt-text, image) pairs from HTML pages (Schuhmann et al., 2022), enabling the recent breakthroughs in Text-to-Image (T2I) modeling. However, replicating this success for videos is limited since a similarly sized (text, video) dataset cannot be easily collected. It would be wasteful to train Text-to-Video (T2V) models from scratch when there already exist models that can generate images. Moreover, unsupervised learning enables networks to learn from orders of magnitude more data. This large quantity of data is important to learn representations of more subtle, less common concepts in the world. Unsupervised learning has long had great success in advancing the field of natural language processing (NLP) (Liu et al., 2019a; Brown et al., 2020). Models pre-trained this way yield considerably higher performance than when solely trained in a supervised manner.

Inspired by these motivations, we propose Make-A-Video. Make-A-Video leverages T2I models to learn the correspondence between text and the visual world, and uses unsupervised learning on unlabeled (unpaired) video data, to learn realistic motion. Together, Make-A-Video generates videos from text without leveraging paired text-video data.

Clearly, text describing images does not capture the entirety of phenomena observed in videos. That said, one can often infer actions and events from static images (e.g. a woman drinking coffee, or an

---

[+] Core Contributors. Corresponding author: urielsinger@meta.com. Jie and Songyang are from University of Rochester (work done during internship at Meta).

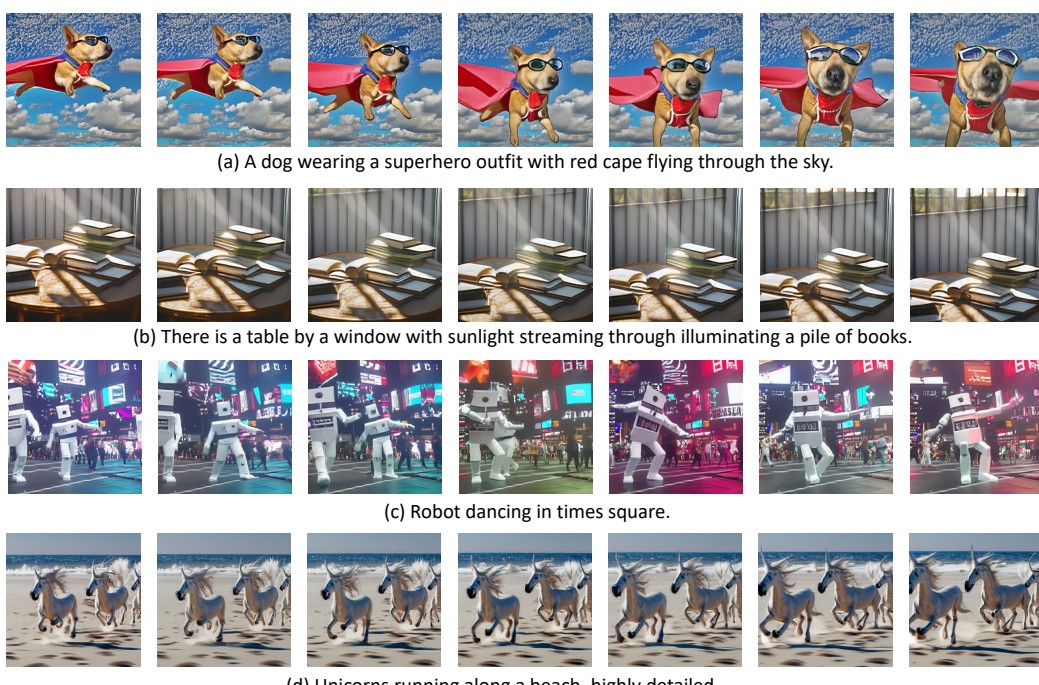

(a) A dog wearing a superhero outfit with red cape flying through the sky.

(b) There is a table by a window with sunlight streaming through illuminating a pile of books.

(c) Robot dancing in times square.

(d) Unicorns running along a beach, highly detailed.

Figure 1: **T2V generation examples.** Our model can generate high-quality videos with coherent motion for a diverse set of visual concepts. In example (a), there are large and realistic motion for the dog. In example (b), the books are almost static but the scene changes with the camera motion. **Video samples are available at make-a-video.github.io**

elephant kicking a football) as done in image-based action recognition systems (Girish et al., 2020). Moreover, even without text descriptions, unsupervised videos are sufficient to learn how different entities in the world move and interact (e.g. the motion of waves at the beach, or of an elephant's trunk). As a result, a model that has only seen text describing images is surprisingly effective at generating short videos, as demonstrated by our temporal diffusion-based method. Make-A-Video sets the new state-of-the-art in T2V generation.

Using function-preserving transformations, we extend the spatial layers at the model initialization stage, to include temporal information. The extended spatial-temporal network includes new attention modules that learn temporal world dynamics from a collection of videos. This procedure significantly accelerates the T2V training process by instantaneously transferring the knowledge from a previously trained T2I network to a new T2V one. To enhance the visual quality, we train spatial super-resolution models as well as frame interpolation models. This increases the resolution of the generated videos, as well as enables a higher (controllable) frame rate.

Our main contributions are:

- We present Make-A-Video – an effective method that extends a diffusion-based T2I model to T2V through a spatiotemporally factorized diffusion model.

- We leverage joint text-image priors to bypass the need for paired text-video data, which in turn allows us to potentially scale to larger quantities of video data.

- We present super-resolution strategies in space and time that, for the first time, generate high-definition, high frame-rate videos given a user-provided textual input.

- We evaluate Make-A-Video against existing T2V systems and present: (a) State-of-the-art results in quantitative as well as qualitative measures, and (b) A more thorough evaluation than existing literature in T2V. We also collect a test set of 300 prompts for zero-shot T2V human evaluation which we plan to release.

## 2 PREVIOUS WORK

**Text-to-Image Generation.** (Reed et al., 2016) is among the first methods to extend unconditional Generative Adversairal Network (GAN) (Goodfellow et al., 2014) to T2I generation. Later GAN variants have focused on progressive generation (Zhang et al., 2017; Hong et al., 2018), or better text-image alignment (Xu et al., 2018; Zhang et al., 2021). The pioneering work of DALL-E (Ramesh et al., 2021) considers T2I generation as a sequence-to-sequence translation problem using a discrete variational auto-encoder (VQVAE) and Transformer (Vaswani et al., 2017). Additional variants (Ding et al., 2022) have been proposed since then. For example, Make-A-Scene (Gafni et al., 2022) explores controllable T2I generation using semantic maps. Parti (Yu et al., 2022a) aims for more diverse content generation through an encoder-decoder architecture and an improved image tokenizer (Yu et al., 2021). On the other hand, Denoising Diffusion Probabilistic Models (DDPMs) (Ho et al., 2020) are successfully leveraged for T2I generation. GLIDE (Nichol et al., 2021) trained a T2I and an upsampling diffusion model for cascade generation. GLIDE's proposed classifier-free guidance has been widely adopted in T2I generation to improve image quality and text faithfulness. DALLE-2 (Ramesh et al., 2022) leverages the CLIP (Radford et al., 2021) latent space and a prior model. VQ-diffusion (Gu et al., 2022) and stable diffusion (Rombach et al., 2022) performs T2I generation in the latent space instead of pixel space to improve efficiency.

**Text-to-Video Generation.** While there is remarkable progress in T2I generation, the progress of T2V generation lags behind largely due to two main reasons: the lack of large-scale datasets with high-quality text-video pairs, and the complexity of modeling higher-dimensional video data. Early works (Mittal et al., 2017; Pan et al., 2017; Marwah et al., 2017; Li et al., 2018; Gupta et al., 2018; Liu et al., 2019b) are mainly focused on video generation in simple domains, such as moving digits or specific human actions. To our knowledge, Sync-DRAW (Mittal et al., 2017) is the first T2V generation approach that leverages a VAE with recurrent attention. (Pan et al., 2017) and (Li et al., 2018) extend GANs from image generation to T2V generation.

More recently, GODIVA (Wu et al., 2021a) is the first to use 2D VQVAE and sparse attention for T2V generation supporting more realistic scenes. NÜWA (Wu et al., 2021b) extends GODIVA, and presents a unified representation for various generation tasks in a multitask learning scheme. To further improve the performance of T2V generation, CogVideo (Hong et al., 2022) is built on top of a frozen CogView-2 (Ding et al., 2022) T2I model by adding additional temporal attention modules. Video Diffusion Models (VDM) (Ho et al., 2022) uses a space-time factorized U-Net with joint image and video data training. While both CogVideo and VDM collected 10M private text-video pairs for training, our work uses solely open-source datasets, making it easier to reproduce.

**Leveraging Image Priors for Video Generation.** Due to the complexity of modeling videos and the challenges in high-quality video data collection, it is natural to consider leveraging image priors for videos to simplifying the learning process. After all, an image is a video with a single frame (Bain et al., 2021). In unconditional video generation, MoCoGAN-HD (Tian et al., 2021) formulates video generation as the task of finding a trajectory in the latent space of a pre-trained and fixed image generation model. In T2V generation, NÜWA (Wu et al., 2021b) combines image and video datasets in a multitask pre-training stage to improve model generalization for fine-tuning. CogVideo (Hong et al., 2022) uses a pre-trained and fixed T2I model for T2V generation with only a small number of trainable parameters to reduce memory usage during training. But the fixed autoencoder and T2I models can be restrictive for T2V generation. The architecture of VDM (Ho et al., 2022) can enable joint image and video generation. However, they sample random independent images from random videos as their source of images, and do not leverage the massive text-image datasets.

Make-A-Video differs from previous works in several aspects. First, our architecture breaks the dependency on text-video pairs for T2V generation. This is a significant advantage compared to prior work, that has to be restricted to narrow domains (Mittal et al., 2017; Gupta et al., 2018; Ge et al., 2022; Hayes et al., 2022), or require large-scale paired text-video data (Hong et al., 2022; Ho et al., 2022). Second, we fine-tune the T2I model for video generation, gaining the advantage of adapting the model weights effectively, compared to freezing the weights as in CogVideo (Hong et al., 2022). Third, motivated from prior work on efficient architectures for video and 3D vision tasks (Ye et al., 2019; Qiu et al., 2017; Xie et al., 2018), our use of pseudo-3D convolution (Qiu et al., 2017) and temporal attention layers not only better leverage a T2I architecture, it also allows for better temporal information fusion compared to VDM (Ho et al., 2022).

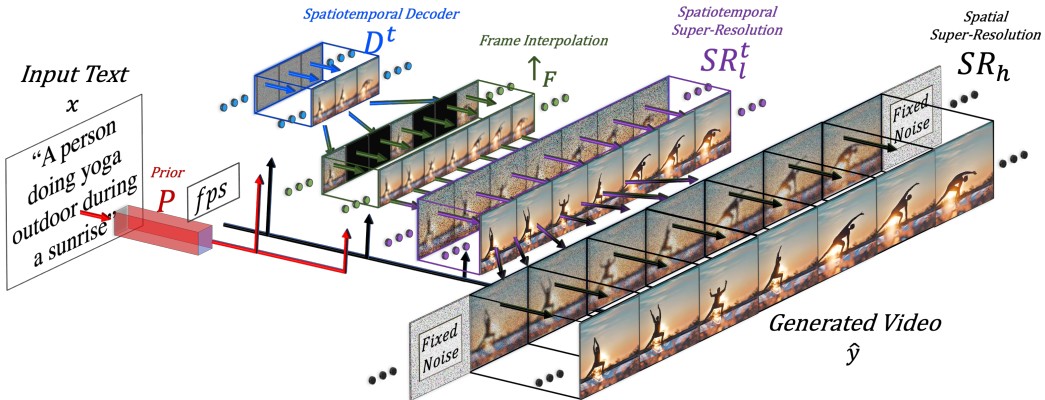

Figure 2: **Make-A-Video high-level architecture.** Given input text $x$ translated by the prior P into an image embedding, and a desired frame rate $fps$, the decoder $\mathrm{D^t}$ generates 16 $64 \times 64$ frames, which are then interpolated to a higher frame rate by $\uparrow_F$, and increased in resolution to $256 \times 256$ by $\mathrm{SR}_l^t$ and $768 \times 768$ by $\mathrm{SR}_h$, resulting in a high-spatiotemporal-resolution generated video $\hat{y}$.

## 3 METHOD

Make-A-Video consists of three main components: **(i)** A base T2I model trained on text-image pairs (Sec. 3.1), **(ii)** spatiotemporal convolution and attention layers that extend the networks' building blocks to the temporal dimension (Sec. 3.2), and **(iii)** spatiotemporal networks that consist of both spatiotemporal layers, as well as another crucial element needed for T2V generation - a frame interpolation network for high frame rate generation (Sec. 3.3).

Make-A-Video's final T2V inference scheme (depicted in Fig. 2) can be formulated as:

$$\hat{y}_t = \mathrm{SR}_h \circ \mathrm{SR}_l^t \circ \uparrow_F \circ \mathrm{D}^t \circ \mathrm{P} \circ (\hat{x}, \mathrm{C}_x(x)), \tag{1}$$

where $\hat{y}_t$ is the generated video, $\mathrm{SR}_h, \mathrm{SR}_l$ are the spatial and spatiotemporal super-resolution networks (Sec. 3.2), $\uparrow_F$ is a frame interpolation network (Sec. 3.3), $\mathrm{D}^t$ is the spatiotemporal decoder (Sec. 3.2), P is the prior (Sec. 3.1), $\hat{x}$ is the BPE-encoded text, $\mathrm{C}_x$ is the CLIP text encoder (Radford et al., 2021), and $x$ is the input text. The three main components are described in detail in the following sections.

### 3.1 TEXT-TO-IMAGE MODEL

Prior to the addition of the temporal components, we train the backbone of our method: a T2I model trained on text-image pairs, sharing the core components with the work of (Ramesh et al., 2022).

We use the following networks to produce high-resolution images from text: **(i) A prior network P**, that during inference generates image embeddings $y_e$ given text embeddings $x_e$ and BPE encoded text tokens $\hat{x}$, **(ii) a decoder network D** that generates a low-resolution $64 \times 64$ RGB image $\hat{y}_l$, conditioned on the image embeddings $y_e$, and **(iii) two super-resolution networks $\mathbf{SR_l,SR_h}$** that increase the generated image $\hat{y}_l$ resolution to $256 \times 256$ and $768 \times 768$ pixels respectively, resulting in the final[1] generated image $\hat{y}$.

### 3.2 SPATIOTEMPORAL LAYERS

In order to expand the two-dimensional (2D) conditional network into the temporal dimension, we modify the two key building blocks that now require not just spatial but also temporal dimensions in order to generate videos: (i) Convolutional layers (Sec. 3.2.1), and (ii) attention layers (Sec. 3.2.2), discussed in the following two subsections. Other layers, such as fully-connected layers, do not require specific handling when adding an additional dimension, as they are agnostic to structured spatial and temporal information. Temporal modifications are made in most U-Net-based diffusion networks: the spatiotemporal decoder $\mathrm{D^t}$ now generating 16 RGB frames, each of size $64 \times 64$, the

---

[1]We then downsample to 512 using bicubic interpolation for a cleaner aesthetic. Maintaining a clean aesthetic for high definition videos is part of future work.

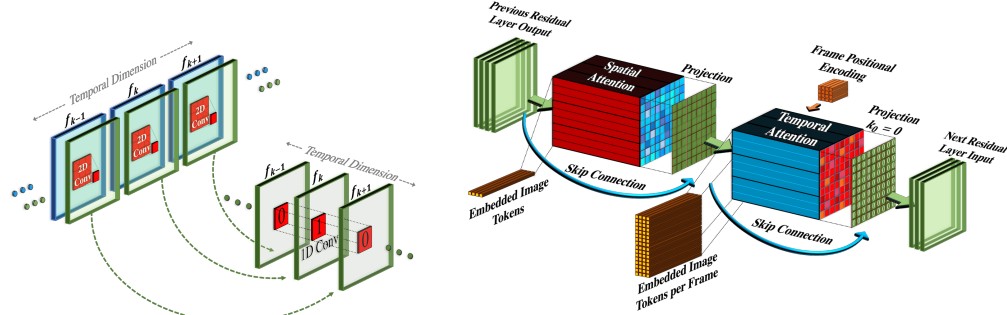

Figure 3: **The architecture and initialization scheme of the Pseudo-3D convolutional and attention layers, enabling the seamless transition of a pre-trained Text-to-Image model to the temporal dimension. (left)** Each spatial 2D conv layer is followed by a temporal 1D conv layer. The temporal conv layer is initialized with an identity function. **(right)** Temporal attention layers are applied following the spatial attention layers by initializing the temporal projection to zero, resulting in an identity function of the temporal attention blocks.

newly added frame interpolation network $\uparrow_F$, increasing the effective frame rate by interpolating between the 16 generated frames (as depicted in Fig. 2), and the super-resolution networks $\mathrm{SR}_l^t$.

Note that super resolution involves hallucinating information. In order to not have flickering artifacts, the hallucination must be consistent across frames. As a result, our $\mathrm{SR}_l^t$ module operates across spatial and temporal dimensions. In qualitative inspection we found this to significantly outperform per-frame super resolution. It is challenging to extend $\mathrm{SR}_h$ to the temporal dimension due to memory and compute constraints, as well as a scarcity of high resolution video data. So $\mathrm{SR}_h$ operates only along the spatial dimensions. But to encourage consistent detail hallucination across frames, we use the same noise initialization for each frame.

### 3.2.1 PSEUDO-3D CONVOLUTIONAL LAYERS

Motivated by separable convolutions (Chollet, 2017), we stack a 1D convolution following each 2D convolutional (conv) layer, as shown in Fig. 3. This facilitates information sharing between the spatial and temporal axes, without succumbing to the heavy computational load of 3D conv layers. In addition, it creates a concrete partition between the pre-trained 2D conv layers and the newly initialized 1D conv layers, allowing us to train the temporal convolutions from scratch, while retaining the previously learned spatial knowledge in the spatial convolutions' weights.

Given an input tensor $h \in \mathbb{R}^{B \times C \times F \times H \times W}$, where $B$, $C$, $F$, $H$, $W$ are the batch, channels, frames, height, and width dimensions respectively, the Pseudo-3D convolutional layer is defined as:

$$Conv_{P3D}(h) := Conv_{1D}(Conv_{2D}(h) \circ T) \circ T, \tag{2}$$

where the transpose operator $\circ T$ swaps between the spatial and temporal dimensions. For smooth initialization, while the $Conv_{2D}$ layer is initialized from the pre-trained T2I model, the $Conv_{1D}$ layer is initialized as the identity function, enabling a seamless transition from training spatial-only layers, to spatiotemporal layers. Note that at initialization, the network will generate K different images (due to random noise), each faithful to the input text but lacking temporal coherence.

### 3.2.2 PSEUDO-3D ATTENTION LAYERS

A crucial component of T2I networks is the attention layer, where in addition to self-attending to extracted features, text information is injected to several network hierarchies, alongside other relevant information, such as the diffusion time-step. While using 3D convolutional layers is computationally heavy, adding the temporal dimension to attention layers is outright infeasible in terms of memory consumption. Inspired by the work of (Ho et al., 2022), we extend our dimension decomposition strategy to attention layers as well. Following each (pre-trained) spatial attention layer, we stack a temporal attention layer, which as with the convolutional layers, approximates a full spatiotemporal attention layer. Specifically, given an input tensor $h$, we define $flatten$ as a matrix operator that

flattens the spatial dimension into $h' \in R^{B \times C \times F \times HW}$. $unflatten$ is defined as the inverse matrix operator. The Pseudo-3D attention layer therefore is therefore defined as:

$$ATTN_{P3D}(h) = unflatten(ATTN_{1D}(ATTN_{2D}(flatten(h)) \circ T) \circ T). \quad (3)$$

Similarly to $Conv_{P3D}$, to allow for smooth spatiotemporal initialization, the $ATTN_{2D}$ layer is initialized from the pre-trained T2I model and the $ATTN_{1D}$ layer is initialized as the identity function.

Factorized space-time attention layers have also been used in VDM (Ho et al., 2022) and CogVideo (Hong et al., 2022). CogVideo has added temporal layers to each (frozen) spatial layers whereas we train them jointly. In order to force their network to train for images and videos interchangeably, VDM has extended their 2D U-Net to 3D through unflattened 1x3x3 convolution filters, such that the subsequent spatial attention remains 2D, and added 1D temporal attention through relative position embeddings. In contrast, we apply an additional 3x1x1 convolution projection (after each 1x3x3) such that the temporal information will also be passed through each convolution layer.

**Frame rate conditioning.** In addition to the T2I conditionings, similar to CogVideo (Hong et al., 2022), we add an additional conditioning parameter $fps$, representing the number of frames-per-second in a generated video. Conditioning on a varying number of frames-per-second, enables an additional augmentation method to tackle the limited volume of available videos at training time, and provides additional control on the generated video at inference time.

**Objectives.** We optimize the model by minimizing the *hybrid* loss following Nichol & Dhariwal (2021); Ramesh et al. (2022) to train the video decoder. Specifically, the loss consists of two terms: a *simple* loss that learns to predict the added noise and a loss $L_{vlb}$ that adds a constraint on the estimated variational lower bound (VLB). The $L_{vlb}$ term is applied the same way as in Nichol & Dhariwal (2021). Thus, we only write the loss term that predicts the added noise as:

$$\mathcal{L}_{decoder} = \mathbb{E}_{C_y(\mathbf{y}_0), \epsilon, fps, t} \left[ \|\epsilon_t - \epsilon_\theta(\mathbf{z}_t, C_y(\mathbf{y}_0), fps, t)\|_2^2 \right] \quad (4)$$

where $\mathbf{y}$ is an input video and $\mathbf{y}_0$ represents the first frame of this video. $C_y(\mathbf{y}_0)$ denotes the extracted CLIP image embedding of the first frame. $\mathbf{z}_t$ is the noisy input added to $\mathbf{y}$ at time step $t$ that is uniformly sampled from 1 to $T$ during training. $fps$ is the frame rate embedding as described above. $\epsilon_t$ is the added noise that is to be estimated by the network represented as $\epsilon_\theta$.

### 3.3 FRAME INTERPOLATION NETWORK

In addition to the spatiotemporal modifications discussed in Sec. 3.2, we train a new masked frame interpolation and extrapolation network $\uparrow_F$, capable of increasing the number of frames of the generated video either by frame interpolation for a smoother generated video, or by pre/post frame extrapolation for extending the video length. In order to increase the frame rate within memory and compute constraints, we fine-tune a spatiotemporal decoder $D^t$ on the task of masked frame interpolation, by zero-padding the masked input frames, enabling video upsampling. When fine-tuning on masked frame interpolation, we add an additional 4 channels to the input of the U-Net: 3 channels for the RGB masked video input and an additional binary channel indicating which frames are masked. We fine-tune with variable frame-skips and $fps$ conditioning to enable multiple temporal upsample rates at inference time. The training objective is the same as Eq. 4 except that we add the additional condition of the unmasked frames. We denote $\uparrow_F$ as the operator that expands the given video tensor through masked frame interpolation. For all of our experiments we applied $\uparrow_F$ with frame skip 5 to upsample a 16 frame video to 76 frames ((16-1)×5+1). Note that we can use the same architecture for video extrapolation or image animation by masking frames at the beginning or end of a video.

### 3.4 TRAINING

The different components of Make-A-Video described above are trained independently. The only component that receives text as input is the prior P. We train it on paired text-image data and do not fine-tune it on videos. The decoder and two super-resolution components are first trained on images alone (no aligned text). Recall that the decoder receives CLIP image embedding as input, and the super-resolution components receive downsampled images as input during training. After training on images, we add and initialize the new temporal layers and fine-tune them over unlabeled video data. 16 frames are sampled from the original video with random $fps$ ranging from 1 to 30. We

Table 1: T2V generation evaluation on MSR-VTT. Zero-Shot means no training is conducted on MSR-VTT. Samples/Input means how many samples are generated (and then ranked) for each input.

| Method | Zero-Shot | Samples/Input | Resolution | FID ($\downarrow$) | CLIPSIM ($\uparrow$) |
|---|---|---|---|---|---|
| GODIVA | No | 30 | $128 \times 128$ | – | 0.2402 |
| NÜWA | No | – | $336 \times 336$ | 47.68 | 0.2439 |
| CogVideo (Chinese) | Yes | 1 | $480 \times 480$ | 24.78 | 0.2614 |
| CogVideo (English) | Yes | 1 | $480 \times 480$ | 23.59 | 0.2631 |
| Make-A-Video (ours) | Yes | 1 | $256 \times 256$ | **13.17** | **0.3049** |

use the beta function for sampling and while training the decoder, start from higher FPS ranges (less motion) and then transition to lower FPS ranges (more motion). The masked-frame-interpolation component is fine-tuned from the temporal decoder.

## 4 EXPERIMENTS

### 4.1 DATASETS AND SETTINGS

**Datasets.** To train the image models, we use a 2.3B subset of the dataset from (Schuhmann et al.) where the text is English. We filter out sample pairs with NSFW images [2], toxic words in the text, or images with a watermark probability larger than $0.5$. We use WebVid-10M (Bain et al., 2021) and a 10M subset from HD-VILA-100M (Xue et al., 2022) [3] to train our video generation models. Note that only the videos (no aligned text) are used. The decoder $D^t$ and the interpolation model is trained on WebVid-10M. $SR_l^t$ is trained on both WebVid-10M and HD-VILA-10M. While prior work (Hong et al., 2022; Ho et al., 2022) have collected private text-video pairs for T2V generation, we use only public datasets (and no paired text for videos). We conduct automatic evaluation on UCF-101 (Soomro et al., 2012) and MSR-VTT (Xu et al., 2016) in a zero-shot setting.

**Automatic Metrics.** For UCF-101, we write one template sentence for each class (without generating any video) and fix it for evaluation. We report Frechet Video Distance (FVD) and Inception Score (IS) on 10K samples following (Ho et al., 2022). We generate samples that follow the same class distribution as the training set. For MSR-VTT, we report Frechet Inception Distance (FID) (Parmar et al., 2022) and CLIPSIM (average CLIP similarity between video frames and text) (Wu et al., 2021a), where all $59,794$ captions from the test set are used, following (Wu et al., 2021b).

**Human Evaluation Set and Metrics.** We collect an evaluation set from Amazon Mechanical Turk (AMT) that consists of 300 prompts. We asked annotators what they would be interested in generating if there were a T2V system. We filtered out prompts that were incomplete (e.g., "jump into water"), too abstract (e.g., "climate change"), or offensive. We then identified 5 categories (animals, fantasy, people, nature and scenes, food and beverage) and selected prompts for these categories. These prompts were selected without generating any videos for them, and were kept fixed. In addition, we also used the DrawBench prompts from Imagen (Saharia et al., 2022) for human evaluation. We evaluate video quality and text-video faithfulness. For video quality, we show two videos in random order and ask annotators which one is of higher quality. For faithfulness, we additionally show the text and ask annotators which video has a better correspondence with the text (we suggest them to ignore quality issues). In addition, we also conducted human evaluation to compare video motion realism of our interpolation model and FILM (Reda et al., 2022). For each comparison, we use the majority vote from 5 different annotators as the final result.

### 4.2 QUANTITATIVE RESULTS

**Automatic Evaluation on MSR-VTT.** In addition to GODIVA and NÜWA that report on MSR-VTT, we also perform inference on the officially released CogVideo model with both Chinese and English inputs for comparison. For CogVideo and Make-A-Video, we only generate one sample for each prompt in a zero-shot setting. We only generate videos that are at $16 \times 256 \times 256$ as the evaluation models do not expect higher resolutions and frame rate. The results are shown in Table 1.

---

[2] We used this model: https://github.com/GantMan/nsfw_model

[3] These 100M clips are sourced from 3.1M videos. We randomly downloaded 3 clips per video to form our HD-VILA-10M subset.

Table 2: Video generation evaluation on UCF-101 for both zero-shot and fine-tuning settings.

| Method | Pretrain | Class | Resolution | IS ($\uparrow$) | FVD ($\downarrow$) |
|---|---|---|---|---|---|
| Zero-Shot Setting | | | | | |
| CogVideo (Chinese) | No | Yes | $480 \times 480$ | 23.55 | 751.34 |
| CogVideo (English) | No | Yes | $480 \times 480$ | 25.27 | 701.59 |
| Make-A-Video (ours) | No | Yes | $256 \times 256$ | **33.00** | **367.23** |
| Finetuning Setting | | | | | |
| TGANv2(Saito et al., 2020) | No | No | $128 \times 128$ | $26.60 \pm 0.47$ | - |
| DIGAN(Yu et al., 2022b) | No | No | | $32.70 \pm 0.35$ | $577 \pm 22$ |
| MoCoGAN-HD(Tian et al., 2021) | No | No | $256 \times 256$ | $33.95 \pm 0.25$ | $700 \pm 24$ |
| CogVideo (Hong et al., 2022) | Yes | Yes | $160 \times 160$ | 50.46 | 626 |
| VDM (Ho et al., 2022) | No | No | $64 \times 64$ | $57.80 \pm 1.3$ | - |
| TATS-base(Ge et al., 2022) | No | Yes | $128 \times 128$ | $79.28 \pm 0.38$ | $278 \pm 11$ |
| Make-A-Video (ours) | Yes | Yes | $256 \times 256$ | **82.55** | **81.25** |

Table 3: Human evaluation results compared to CogVideo (Hong et al., 2022) on DrawBench and our test set, and to VDM (Ho et al., 2022) on the 28 examples from their website. The numbers show the percentage of raters that prefer the results of our Make-A-Video model.

| Comparison | Benchmark | Quality | Faithfulness |
|---|---|---|---|
| Make-A-Video (ours) vs. VDM | VDM prompts (28) | 84.38 | 78.13 |
| Make-A-Video (ours) vs. CogVideo (Chinese) | DrawBench (200) | 76.88 | 73.37 |
| Make-A-Video (ours) vs. CogVideo (English) | DrawBench (200) | 74.48 | 68.75 |
| Make-A-Video (ours) vs. CogVideo (Chinese) | Our Eval. Set (300) | 73.44 | 75.74 |
| Make-A-Video (ours) vs. CogVideo (English) | Our Eval. Set (300) | 77.15 | 71.19 |

Make-A-Video's zero-shot performance is much better than GODIVA and NÜWA which are trained on MSR-VTT. We also outperform CogVideo in both Chinese and English settings. Thus, Make-A-Video has significantly better generalization capabilities than prior work.

**Automatic Evaluation on UCF-101.** UCF-101 is a popular benchmark to evaluate video generation and has been recently used in T2V models. CogVideo performed finetuning of their pretrained model for class-conditional video generation. VDM (Ho et al., 2022) performed unconditional video generation and trained from scratch on UCF-101. We argue that both settings are not ideal and is not a direct evaluation of the T2V generation capabilities. Moreover, the FVD evaluation model expects the videos to be $0.5$ second (16 frames), which is too short to be used for video generation in practice. Nevertheless, in order to compare to prior work, we conducted evaluation on UCF-101 in both zero-shot and finetuning settings. As shown in Table 2, Make-A-Video's zero-shot performance is already competitive than other approaches that are trained on UCF-101, and is much better than CogVideo, which indicates that Make-A-Video can generalize better even to such a specific domain. Our finetuning setting achieves state-of-the-art results with a significant reduction in FVD, which suggests that Make-A-Video can generate more coherent videos than prior work.

**Human Evaluation.** We compare to CogVideo (the only public zero-shot T2V generation model) on DrawBench and our test set. We also evaluate on the 28 videos shown on the webpage of VDM (Ho et al., 2022) (which may be biased towards showcasing the model's strengths). Since this is a very small test set, we randomly generate 8 videos for each input and perform evaluation 8 times and report the average results. We generate videos at $76 \times 256 \times 256$ resolution for human evaluation. For VDM, it is worth noting that we have achieved significantly better results The results are shown in Table 3. Make-A-Video achieves much better performance in both video quality and text-video faithfulness in all benchmarks and comparisons. For CogVideo, the results are similar on DrawBench and our evaluation set. Additional experiments combining components of CogVideo & Make-A-Video in order to measure the efficacy of different components are presented in Sec. 6.1. without any cherry-picking. We also evaluate our frame interpolation network in comparison to

FILM (Reda et al., 2022). We first generate low frame rate videos (1 FPS) from text prompts in DrawBench and our evaluation set, then use each method to upsample to 4 FPS. Raters choose our method for more realistic motion 62% of the time on our evaluation set and 54% of the time on DrawBench. We observe that our method excels when there are large differences between frames where having real-world knowledge of how objects move is crucial.

### 4.3 QUALITATIVE RESULTS

Examples of Make-A-Video's generations are shown in Figure 1. In this section, we will show T2V generation comparison to CogVideo (Hong et al., 2022) and VDM (Ho et al., 2022), and video interpolation comparison to FILM (Reda et al., 2022). In addition, our models can be used for a variety of other tasks such as image animation, video variation, etc. Due to space constraint, we only show a single example of each. Figure 4 (a) shows the comparison of Make-A-Video to CogVideo and VDM. Make-A-Video can generate richer content with motion consistency and text correspondence. Figure 4 (b) shows an example of image animation where we condition the masked frame interpolation and extrapolation network $\uparrow_F$ on the image and CLIP image embedding to extrapolate the rest of the video. This allows a user to generate a video using their own image – giving them the opportunity to personalize and directly control the generated video. Figure 4 (c) shows a comparison of our approach to FILM (Reda et al., 2022) on the task of interpolation between two images. We achieve this by using the interpolation model that takes the two images as the beginning and end frames and masks 14 frames in between for generation. Our model generates more semantically meaningful interpolation while FILM seems to primarily smoothly transition between frames without semantic real-world understanding of what is moving. Figure 4 (d) shows an example for video variation. We take the average CLIP embedding of all frames from a video as the condition to generate a semantically similar video. More video generation examples and applications can be found here: make-a-video.github.io.

## 5 DISCUSSION

Learning from the world around us is one of the greatest strengths of human intelligence. Just as we quickly learn to recognize people, places, things, and actions through observation, generative systems will be more creative and useful if they can mimic the way humans learn. Learning world dynamics from orders of magnitude more videos using unsupervised learning helps researchers break away from the reliance on labeled data. The presented work has shown how labeled images combined effectively with unlabeled video footage can achieve that.

As a next step we plan to address several of the technical limitations. As discussed earlier, our approach can not learn associations between text and phenomenon that can only be inferred in videos. How to incorporate these (e.g., generating a video of a person waving their hand left-to-right or right-to-left), along with generating longer videos, with multiple scenes and events, depicting more detailed stories, is left for future work.

Also, the lack of standard benchmarks in the field of large-scale generative models makes it difficult for works to compare with each other and measure progress over time. To address this issue, we went beyond what was done in most existing works, with extensive human evaluation, including a comparison to examples shared by authors of existing approaches (or generated using models when publicly released). We hope the community will continue making progress towards better benchmarks for generative models.

As with all large-scale models trained on data from the web, our models have learnt and likely exaggerated social biases, including harmful ones. Our T2I generation model was trained on data that removed NSFW content and toxic words. All our data (image as well as videos) is publicly available, adding a layer of transparency to our models, and making it possible for the community to reproduce our work.

ACKNOWLEDGMENTS

Mustafa Said Mehmetoglu, Jacob Xu, Katayoun Zand, Jia-Bin-Huang, Jiebo Luo, Shelly Sheynin, Angela Fan, Kelly Freed. Thank you for your contributions! Thank you as well to all the people internal to FAIR who helped enable this work by providing extra compute for our experimentation.

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

# 6 APPENDIX

## 6.1 DISENTANGLING EFFICACY OF THE T2I AND I2V COMPONENTS

We evaluate two additional baselines to disentangle the contributions of two components of Make-A-Video – Text-to-Image (T2I) and Image-to-Video (I2V). We do so by using Make-A-Video's T2I with CogVideo's I2V, and analogously CogVideo's T2I with Make-A-Video's I2V.

Note that Make-A-Video does not have two explicit T2I and I2V modules. That is, it does not generate an image first and then use it's CLIP embedding to generate the 16 video frames. It generates the 16 video frames directly from the image CLIP embedding which is predicted using the prior. As a result, in order to evaluate an ablation like this, we first generate a frame with CogVideo's T2I module, extract the image CLIP embedding from it, and use that to generate the 16 video frames.

We perform our ablation in a zero-shot setting on the MSR-VTT dataset where we generate 5K samples using each of the approaches. We condition CogVideo on English because English prompts performed better in our MSR-VTT evaluation (see Tab. 1).

Please see Tab. 4 for the results of this ablation study. We report FID and CLIPSIM metrics computed on static frames generated by the approaches. In addition, to evaluate temporal quality, we perform human evaluation on 200 videos out of the generated 5K. Each video was rated by 5 human evaluators, and we take the majority vote as the final result. We report the percentage of raters that prefer the results of our Make-A-Video model over the two baselines.

We see that Make-A-Video is favored across all metrics when compared to two of its CogVideo variants. Human evaluators preferred generations that used Make-A-Video's spatial-temporal mechanism (I2V) over CogVideo's about 2 out of 3 times. Similarly, evaluators preferred generations using Make-A-Video's T2I module over CogVideo's 3 out of 4 times. We also report CogVideo performance (CogVideo T2I and I2V) as reference.

Table 4: Evaluating the contribution of the T2I and I2V components in zero-shot generation on MSR-VTT. We report FID and CLIPSIM scores computed on static frames. Quality shows the percentage of human raters that prefer the results of our Make-A-Video model over the baselines.

| Method | FID ($\downarrow$) | CLIPSIM ($\uparrow$) | Quality ($\uparrow$) |
|---|---|---|---|
| CogVideo (English) | 20.01 | 0.201 | 60% |
| CogVideo T2I + Make-A-Video I2V | 18.42 | 0.251 | 74% |
| Make-A-Video T2I + CogVideo I2V | 14.09 | 0.302 | 66% |
| Make-A-Video (ours) | **13.96** | **0.305** | – |

## 6.2 ABLATION STUDY

We perform an ablation study on several architecture and training choices. First, we ablate our architecture design. Specifically the contribution of temporal convolutional layers (Sec. 3.2.1) and temporal attention layers (Sec. 3.2.2). Second, we demonstrate the effectiveness of initializing our Text-to-Video model with pre-trained Text-to-Image model weights.

We perform our ablation in a zero-shot setting on the MSR-VTT dataset where we generated videos for 6K sentences. We report the automatic CLIPSIM metric to evaluate text faithfullness. In addition, to evaluate temporal quality, we generate videos for our human evaluation set of 300 prompts and ask raters to select which model's generation is higher quality. Each pair of videos was rated by 5 human evaluators, and we take the majority vote as the final result. We report the percentage of raters that prefer the results of our Make-A-Video model over the two baselines. Please see Tab. 5 for the results of this ablation study.

In our architecture design ablation we trained two variants: (i) Make-A-Video architecture without temporal attention layers with the spatial attention layers kept as is - "No Attn", (ii) Make-A-Video architecture without temporal convolutional layers with the spatial convolutional layers kept as is - "No Conv", and, (iii) our complete Make-A-Video architecture - "Full". All models were trained

Table 5: Ablation study on architecture and training design choices. Results are reported on zero-shot generation on MSR-VTT. We report CLIPSIM scores computed on static frames. Quality shows the percentage of human raters that prefer the results of our Make-A-Video model over the baselines.

| Method | CLIPSIM ($\uparrow$) | Quality ($\uparrow$) |
|---|---|---|
| From Scratch | 0.246 | 63.31% |
| No Conv | 0.256 | 52.04% |
| No Attn | 0.257 | 55.25% |
| Full | 0.258 | – |

Table 6: Human evaluation comparing the effects of different components. Results are evaluated on the human evaluation set with 300 prompts. Quality shows the percentage of the human raters who prefer the results of setting B in each comparison.

| Comparison | Setting A | Setting B | Quality ($\uparrow$) |
|---|---|---|---|
| 1 | $16 \times 64 \times 64$ | $16 \times 256 \times 256$ | 92.48% |
| 2 | $16 \times 256 \times 256$ | $76 \times 256 \times 256$ | 68.30% |
| 3 | $16 \times 256 \times 256$ | $16 \times 768 \times 768$ | 60.13% |
| 4 | $16 \times 256 \times 256$ (static SR) | $16 \times 256 \times 256$ | 54.25% |
| 5 | $16 \times 768 \times 768$ (random noise) | $16 \times 768 \times 768$ | 50.98% |

for 100K iterations. As can be seen, both the temporal convolutional layers and temporal attention layers are important to improve video quality and text faithfulness.

In addition to architecture design ablation, we justify our decision to initialize the T2V model with the weights of a pre-trained T2I model. We begin by reporting CLIPSIM and subjective quality evaluation metrics on a T2V model trained from scratch 100K iterations - "From Scratch". As can be seen in Tab. 5, the model initialized with a pre-trained T2I weights ("Full") outperforms the model trained from scratch when trained the same number of iterations. In addition, the model initialized with pre-trained T2I weights achieves the CLIPSIM score of the model trained from scratch after just 50K iterations, demonstrating the acceleration achieved by initializing with the weights of a T2I model.

## 6.3 EFFECTS OF DIFFERENT COMPONENTS

Our framework consists of several components that are independently trained and sequentially applied during inference. These models include: 1) a decoder that generates a video of $16 \times 64 \times 64$ from the image embedding generated from a prior model; 2) an interpolation model that improves the frame rate and generates a video of $76 \times 64 \times 64$; 3) a temporal super-resolution model that improves the video resolution by considering temporal information and generates a video of $76 \times 256 \times 256$; 4) a second super-resolution model that is applied independently on each frame with the same sampled frame noise and generates the final video of $76 \times 768 \times 768$.

We study the contributions of each of these components through human evaluation. The results are shown in Table 6. We have the following observations. First, improving the resolution from 64 to 256 helps boost the video quality significantly (Comparison 1). Second, increasing the frame rate from 4 fps (16 frames) to 19 fps (76 frames) also help to increase the quality quite a bit (Comparison 2). Third, further increasing the resolution from 256 to 768 can still boost the quality (Comparison 3). These three comparisons have demonstrated the effectiveness of our interpolation model, and two super-resolution models in improving the quality of the generated videos.

Furthermore, we also compare our temporal super-resolution model with a static super-resolution model. The latter is applied independently on each frame without considering temporal information. As shown in Table 6 (Comparison 4), the temporal super-resolution model shows better video quality compared to the static super-resolution model. This justifies our use of a temporal super-resolution model at the 256 resolution level. Another comparison we have done is to validate the effect of a fixed frame noise for the second super-resolution model. As shown in the last row of 6, using fixed noise has a slightly better result compared to using random noise for each frame.

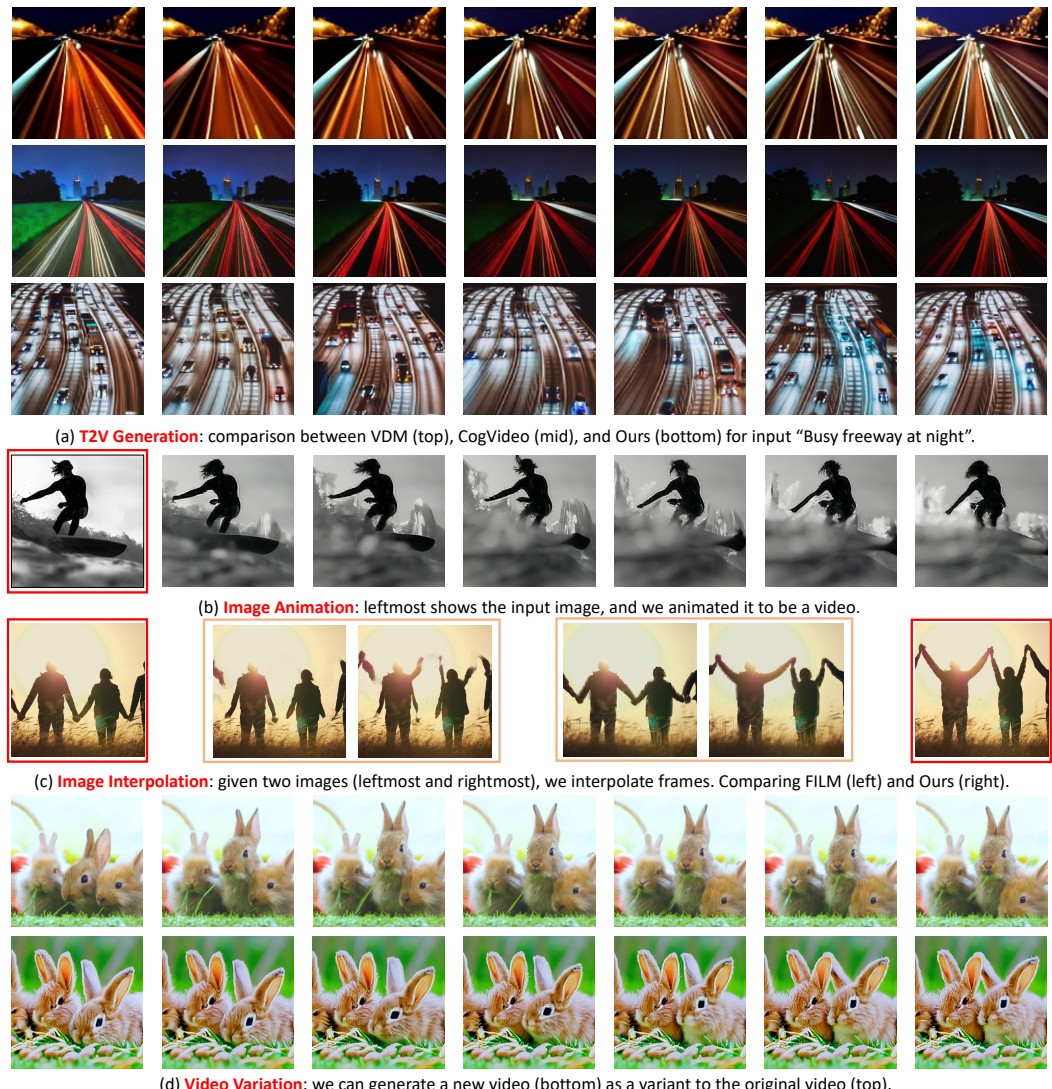

(a) **T2V Generation**: comparison between VDM (top), CogVideo (mid), and Ours (bottom) for input "Busy freeway at night".

(b) **Image Animation**: leftmost shows the input image, and we animated it to be a video.

(c) **Image Interpolation**: given two images (leftmost and rightmost), we interpolate frames. Comparing FILM (left) and Ours (right).

(d) **Video Variation**: we can generate a new video (bottom) as a variant to the original video (top).

Figure 4: Qualitative results for various comparisons and applications.

Table 7: Hyperparameters for the models

| | $P$ | $D\,;D^t$ | $\uparrow_F$ | $SR_l\,;SR_l^t$ | $SR_h$ |
|---|---|---|---|---|---|
| Diffusion steps | 1000 | 1000 | 1000 | 1000 | 1000 |
| Noise schedule | cosine | cosine | cosine | cosine | linear |
| Objective | $x_{start}$ | $\epsilon$ | $\epsilon$ | $x_{start}$ | $x_{start}$ |
| Sampling steps | 64 | 100 | 50 | 50 | 50 |
| Sampling variance method | analytic | DDPM | DDPM | DDIM | DDIM |
| Crop fraction | - | - | - | $\frac{1}{2}$ | $\frac{1}{3}$ |
| Model size | $1.3B$ | $2.2B\,;3.1B$ | $3.1B$ | $1B\,;1.4B$ | $730M$ |
| Channels | - | 512 | 512 | 320 | 320 |
| Depth | - | 3 | 3 | 3 | 3 |
| Channels multiple | 64 | $1,2,3,4$ | $1,2,3,4$ | $1,1,2,2,4,4$ | $1,2,3,4$ |
| Heads channels | - | 64 | 64 | - | - |
| Attention resolution | - | $32,16,8$ | $32,16,8$ | - | - |
| Text encoder context | 128 | - | - | - | - |
| Text encoder width | 2048 | - | - | - | - |
| Text encoder depth | 24 | - | - | - | - |
| Text encoder heads | 32 | - | - | - | - |
| Dropout | - | 0.1 | 0.1 | 0.1 | 0.1 |
| Weight decay | $6.0e-2$ | - | - | - | - |
| Batch size | 4096 | 2048 ; 512 | 512 | 1024 ; 256 | 1024 |
| Iterations | $3M$ | $2M\,;200K$ | $100K$ | $700K\,;150K$ | $700K$ |
| Learning rate | $1.1e-4$ | $6.0e-5$ | $6.0e-5$ | $1.2e-4\,;6.0e-5$ | $1.2e-4$ |
| Adam $\beta_2$ | 0.96 | 0.999 | 0.999 | 0.999 | 0.999 |
| Adam $\epsilon$ | $1.0e-6$ | $1.0e-8$ | $1.0e-8$ | $1.0e-8$ | $1.0e-8$ |
| EMA decay | 0.9999 | 0.9999 | 0.9999 | 0.9999 | 0.9999 |
| Model Parameters (B) | 1.3 | 3.1 | 3.1 | 1.4 | 0.7 |

