# OpenReview forum: "Make-A-Video: Text-to-Video Generation without Text-Video Data"
_ICLR.cc/2023/Conference — ICLR 2023 poster_

### Official Review · Reviewer_5k2D · 2022-10-25

**Confidence:** 4
**Correctness:** 3
**Technical Novelty And Significance:** 3
**Empirical Novelty And Significance:** 3
**Recommendation:** 6

**Clarity, Quality, Novelty And Reproducibility:**

This work uses solely open-source datasets, making it easier to reproduce. Will the code and model be released?

**Strength And Weaknesses:**

**Strength**

1) This paper is well-written and easy to follow.

2) The method is simple but effective.

3) The results are compelling.

4) This work uses solely open-source datasets, making it easier to reproduce.

**Weakness**

1) A relatively comprehensive benchmarks. The automatic quantitative evaluation is only done in MSR-VTT and UCF-101. The class and coverage is somewhat limited. Since the results on general situation is more for fun rather than the practical application, some domain-specific evaluation is wanted, to evaluate the practical application ability of the proposed models. For example, text2face, text2human or text2scene, which could has broad application scenarios. Thus, a more comprehensive benchmarks, especially for domain-specific evaluations, is wanted.

2) A absolutely fair comparison to SOTA. For the results on Table 1 and 2, the resolution, number of parameters are different for these compared models. It makes the comparison with SOTA of IS and FID seem to be weak, since other factors are neglected. A absolutely fair comparison benchmark is also wanted.

3) The precise control of action in sequence. It will be better if authors can show the results for text2face/text2human, which is more challenging than general situations, since the more subtle action should be generated. Could the model control the action precisely, for example, with the text: a young man first smiles, then nod his head and finally turns left to speak to others. Will the subtle action and the sequential relation be faithfully maintained?


**Summary Of The Paper:**

This work presents a text2video method, which builds on image2video model with spatial-temporal modules.
The method is straight-forward and the results are appealing.

**Summary Of The Review:**

This work presents a new method for text to video generation, which is a really important problem. The method is effective and the results are compelling. In general, I like this paper and recommend for acceptance so far.

The main concerns are listed in Weakness above. Although some of the concerns are not specific for this paper but for this research domain, which lacks standard benchmark, It will be great to see the opinions and discussions for this problem in this paper.

Many previous methods are more likely to show off the results rather than make it as a research problem. This paper tried to use open-source datasets, which has started to consider the reproducibility for the community. I really appreciate this awareness.

---

> ### Author Response · Authors · 2022-11-14
> **Authors response to reviewer 5k2D**
>
> We acknowledge the reviewer’s concerns – although partially not specific to our paper – about the lack of standard benchmarking in this domain. We agree with this observation, and added a paragraph discussing this issue to the Discussion section.
>
> > A relatively comprehensive benchmarks. The automatic quantitative evaluation is only done in MSR-VTT and UCF-101. The class and coverage is somewhat limited. Since the results on general situation is more for fun rather than the practical application, some domain-specific evaluation is wanted, to evaluate the practical application ability of the proposed models. For example, text2face, text2human or text2scene, which could has broad application scenarios. Thus, a more comprehensive benchmarks, especially for domain-specific evaluations, is wanted.
>
> We agree. More specific application-centric benchmarks will help address these concerns.
>
> > An absolutely fair comparison to SOTA. For the results on Table 1 and 2, the resolution, number of parameters are different for these compared models. It makes the comparison with SOTA of IS and FID seem to be weak, since other factors are neglected. A absolutely fair comparison benchmark is also wanted.
>
> Following the reviewer’s comment, we have added the resolution to the MSR-VTT evaluation tables. We agree an absolutely fair comparison benchmark is needed. But unfortunately, the fact that all prior work has used different resolutions is out of our control. We have done our best to compare with them as fair as possible. For example, the CogVideo paper reports that they can generate videos of 160x160 resolution. In their demo, they added the super-resolution model from CogView2 to increase the resolution to 480x480. Since 160x160 is lower than our resolution, we compare our generation at 256x256 to the 480x480 resolution of CogVideo in all the evaluations when we use their demo to generate the videos. Despite this, we outperform CogVideo on all datasets.
>
> > The precise control of action in sequence. It will be better if authors can show the results for text2face/text2human, which is more challenging than general situations, since the more subtle action should be generated. Could the model control the action precisely, for example, with the text: a young man first smiles, then nod his head and finally turns left to speak to others. Will the subtle action and the sequential relation be faithfully maintained?
>
> We agree with the reviewer’s observation. We’ve indicated in the discussion section: “As a next step we plan to address several of the technical limitations. As discussed earlier, our approach can not learn associations between text and phenomenon that can only be inferred in videos. How to incorporate these (e.g., generating a video of a person waving their hand left-to-right or right-to-left), along with generating longer videos, with multiple scenes and events, depicting more detailed stories, is left for future work (as also mentioned in the Discussion section of the submitted paper).”

---

### Official Review · Reviewer_hyJV · 2022-10-26

**Confidence:** 5
**Correctness:** 3
**Technical Novelty And Significance:** 3
**Empirical Novelty And Significance:** 2
**Recommendation:** 6

**Clarity, Quality, Novelty And Reproducibility:**

- Clarity: The paper is clear and easy to read, however substantial details are missing
- Quality: The paper achieved some high quality results which can push the field forward.
- Novelty: The main idea (learning and applying dynamics on top of T2I systems) is interesting and novel.
- Reproducibility: The paper gets poor score on reproduciblility. There are many missing details and the computational requirements are omitted.

**Strength And Weaknesses:**

===== Strengths
+ The main idea behind the paper is well motivated and an interesting one. There are no massive dataset of labeled video datasets (yet!) while it is relatively easier to collect large amount of raw videos.
+ The paper is easy to read and understand. The notations are clear to follow which makes it easier to follow what it going on.
+ There are many qualitative examples on the appended website: https://gen-videos.github.io/ This is always a big plus for video papers.
+ While the quantitative benchmarks for video generation are quite weak, authors did their best to compare with previous work in a fair setting.

===== Weaknesses
- Missing details. Unfortunately, there are *many* missing details in the paper which substantially reduces its impact on the field. The biggest one imho is the loss. While the system consists of many smaller parts, trained separately, it is not clearly what the loss for any of these components are. In fact, the term "loss" or "objective" which the authors heavily relied on to optimize, does not appear in the paper even once! This is really interesting for a deep learning paper. While I understand that creating such systems (unfortunately) is limited to big industrial labs, reproducibility still should be a big concern.

- Quantitative results are exaggerated. It is no secret that the benchmarks for video generation are still quite weak. The metrics (e.g. FVD) are inconclusive and unrepresentative of the differences. While I appreciate the authors effort to provide a large set of videos in their website, as well as a short human eval, it seems the quantitative results in Table 1 and 2 are exaggerated and in absence of evaluation details it is hard to check that clearly.  Particularly, FVD is sensitive to resolution but this paper compares different resolution without mentioning how.  but how? Higher resolution usually results in better FVD (because of I3D resolution)  and therefore comparing Make-A-Video at 256x256 to e.g. CogVideo at 160x160 or VMD at 64x64 is not a fair comparison. Also for CogVideo, it is not clear where the videos came from, because the official published version is only in Chinese and as of writing this review, the official github page clearly mentions that "Currently only Chinese input is supported." To make things even worse, the provided CogVideo videos in your website,  are significantly worse compared to CogVideo's official page. Again, missing details here severely affects the quality of the paper.

- Lack of analysis. The main idea of learning dynamics from a large corpus of videos and then applying them to generated images is interesting and novel to the best of my knowledge. However, like any other system with multiple components, it has clear down sides compare to an e2e system. First of all, the dataset of videos that the authors used (WebVid) has labels. The paper clearly mentions that it did not use these labels but this raises the question of why? The paper does not provide any analysis on how using more videos improves the system and if 10M labeled videos was enough to learn dynamics then the motivation of the paper is questionable. Second, the paper does not provide any analysis on the weaknesses of the proposed method. I would assume a system that learns the dynamics separately will have weak correlation between dynamics and the terms to describe them (e.g. walking jumping dancing etc), is this the case? if not why not? Where does the alignment come from? Or as another example, how long the videos can be before becoming incoherent? The website includes a few examples of "long" videos however it's not clear how far the model can be pushed and what its limitations are. In absence of any such analysis and the fact the code/model is behind tall walls, it is hard to answer such questions.

**Summary Of The Paper:**

The paper proposes Make-A-Video, an approach for generating short video clips conditioned on a given open domain text. The model does not need any text-video pairs and instead relies on learning dynamics from unlabeled videos and applying them to existing text to image systems. The approach also applies multiple levels of super resolution, both temporal and spatial, to increase the resolution of generated videos.

**Summary Of The Review:**

Overall, while the results of the paper is impressive, it lacks substantial details which makes it close to impossible to reproduce, no analysis to judge the limitations of the approach and exaggerated quantitative numbers. Therefore, the paper should be improved, scientifically, to be acceptable at a major scientific conference, imho. Particularly, more details, more analysis and fairer comparison.

---

> ### Author Response · Authors · 2022-11-14
> **Authors response to reviewer hyJV [1/2]**
>
> We are confused by the review. On one hand, the review indicates that one of the strengths of this work is that the “authors did their best to compare with previous work in a fair setting”, however it later indicates as weaknesses “Lack of analysis” and “Quantitative results are exaggerated”. The comparison to CogVideo was done with the publicly shared model and samples at the time of submission.
> Crucially though, our comparisons to Video Diffusion Models (VDM) should reinforce the above that, even with the selected samples the authors of VDM published on their website, we compare to them favorably, both qualitatively as well as quantitatively.
>
> > “While I understand that creating such systems (unfortunately) is limited to big industrial labs, reproducibility still should be a big concern.”
>
> Despite the non trivial compute requirement, we agree with 5k2D that: “this work uses solely open-source datasets, making it easier to reproduce.”. We also agree with X9uC that “considering the simplicity of the method, the reproducibility should be high.”. We also want to note that “big industrial labs” is not relevant to reviewing the contributions of this paper.
>
>
> > “Quantitative results are exaggerated. It is no secret that the benchmarks for video generation are still quite weak .. The metrics (e.g. FVD) are inconclusive and unrepresentative of the differences.”
>
> While we agree that the mentioned metrics are not perfect, it does not mean that our reported metrics are “exaggerated”. Note that it is due to the imperfect nature of these automatic metrics that we additionally perform extensive human evaluation.
>
> > “Higher resolution usually results in better FVD (because of I3D resolution) and therefore comparing Make-A-Video at 256x256 to e.g. CogVideo at 160x160 or VMD at 64x64 is not a fair comparison”
>
> Prior work has used different settings (resolution varies). For the finetuning setting, we can only cite these numbers as prior works did not release their trained models on UCF. We did compare to CogVideo on 480x480 for zero-shot setting on UCF. Moreover, we compared our 256x256 generation to CogVideo’s 480x480 generation for all zero-shot evaluation and human evaluation. For CogVideo, we did the inference using the official released code.
>
> > “for CogVideo, it is not clear where the videos came from, because the official published version is only in Chinese and as of writing this review, the official github page clearly mentions that "Currently only Chinese input is supported." To make things even worse, the provided CogVideo videos in your website, are significantly worse compared to CogVideo's official page. Again, missing details here severely affects the quality of the paper.”
>
> Thank you for this suggestion. With respect to the comparison to CogVideo, we have conducted an additional disentangling efficacy study that perhaps sheds more light on the contributions of each of the components when replaced with the other. Please refer to Section 6.1 of the updated revision.
>
> Although CogVideo is trained on Chinese only, it is based on a frozen CogView-2, which was trained on both Chinese and English. So we used both English and Chinese for a more comprehensive evaluation. On CogVideo results, we used their model/code as is for the generation and did nothing to make the quality worse, we suggest the reviewer to check the demo to assess the quality in general (compared to potentially cherry-picked results on the website): https://models.aminer.cn/cogvideo/.
>
> > “it is not clearly what the loss for any of these components…  This is really interesting for a deep learning paper “
>
> Thank you for this feedback.  We have added the training objective which we optimize for in the updated revision (Eq. 4). We have also added a table of all hyperparameters in the appendix (Table 7).

---

> > ### Author Response · Authors · 2022-11-14
> > **Authors response to reviewer hyJV [2/2]**
> >
> > > “Lack of analysis. The main idea of learning dynamics from a large corpus of videos and then applying them to generated images is interesting and novel to the best of my knowledge. However, like any other system with multiple components, it has clear down sides compare to an e2e system. First of all, the dataset of videos that the authors used (WebVid) has labels. The paper clearly mentions that it did not use these labels but this raises the question of why? The paper does not provide any analysis on how using more videos improves the system and if 10M labeled videos was enough to learn dynamics then the motivation of the paper is questionable. Second, the paper does not provide any analysis on the weaknesses of the proposed method. I would assume a system that learns the dynamics separately will have weak correlation between dynamics and the terms to describe them (e.g. walking jumping dancing etc), is this the case? if not why not? Where does the alignment come from? Or as another example, how long the videos can be before becoming incoherent? The website includes a few examples of "long" videos however it's not clear how far the model can be pushed and what its limitations are. In absence of any such analysis and the fact the code/model is behind tall walls, it is hard to answer such questions.“
> >
> >
> > We discuss the limitations of our approach in the Discussion section of the submitted version of the paper: “our approach can not learn associations between text and phenomenon that can only be inferred in videos. How to incorporate these (e.g., generating a video of a person waving their hand left-to-right or right-to-left), along with generating longer videos, with multiple scenes and events, depicting more detailed stories, is left for future work. “.  On video duration, while we have added some longer video generations on the website (as hyJV notes), we found that it is challenging to generate consistently coherent videos longer than 5 seconds (and is correspondingly noted as part of future work in the Discussion section, also quoted above).
> >
> > The ablation studies in Sections 6.1 and 6.2 of the revised version sheds light on the role of the individual components of our approach, including the value of initializing the model with a text-to-image generation model. Note that we do compare to existing e2e systems such as CogVideo and Video Diffusion Models.
> >
> > We did not use the labels in WebVid to assess feasibility of training a video generation model without text-video paired data which is difficult to gather. This is analogous to works studying self supervised learning on ImageNet even though ImageNet has labels. With the Make-A-Video results, the community can now take the next step of potentially training with a larger number of (unlabeled) videos to see if that improves generation quality further

---

> > > ### Comment · Reviewer_hyJV · 2022-11-20
> > > **Thank you for additional details. The paper still has issues.**
> > >
> > > Thank you for the response. Unfortunately, your rebuttal does not address my main concern. I increased my rating slightly given the newly added details and experiments but the paper still has major issues. Please find more details below.
> > >
> > > >> We are confused by the review.
> > >
> > > "did your best" doesn't mean it's perfect. I simply applauded the effort and pointed out the issues. How's that confusing?
> > >
> > > >> We also want to note that “big industrial labs” is not relevant to reviewing the contributions of this paper.
> > >
> > > I pointed out the paper being from "big industry" to encourage you to put more effort for making your paper more reproducible for "outside", otherwise I agree that there is no point. I strongly disagree with other reviewers that found this paper reproducible given that it didn't even mention its training objective or any hyper-parameters ¯\_(ツ)_/¯ I agree that overall method is described in a way which "feels" reproducible but the devil is in the details. That being said, I appreciate your efforts in the the updated version is clearly better in these terms and this is why I increased my recommendation.
> > >
> > > >> On CogVideo results, we used their model/code as is for the generation and did nothing to make the quality worse
> > >
> > > To be clear, I was not suggesting any malicious activity and I'm sorry if my suggestion could be read that way. However, comparing the CogVideo videos on your website and the their website, there is a clear quality gap. It can be justified by the cherry-picked nature of the original website, or, the fact that their website is a translated version from Chinese and the model just doesn't work great for English (as CLEARLY indicated on their github page). So maybe any non-chinese comparison is not fair? (I understand the origin of CogView but that still doesn't justify the gap.) I encourage you to communicate with the CogVideo team to see what they think about such comparison.
> > > Please note that conducting human experiments is not going to solve this issue since the problem is the videos themselves.
> > >
> > > >>  we found that it is challenging to generate consistently coherent videos longer than 5 seconds.
> > >
> > > Please consider adding failed examples and more details about the limitations. The same helps with other arguments in the Discussion section such as text-video correlation.
> > >
> > > >> We did not use the labels in WebVid to assess feasibility of training a video generation model without text-video paired data which is difficult to gather.
> > >
> > > My main argument is that the motivation here is ill stated. If about 10M videos (without label) is good enough for training a video model (which clearly is, given this paper!) AND labeling this many videos is not too difficult (as done by WebVid), then why not use the labels? I agree that not having text-video labels help with "scaling" to larger datasets but this paper does not show that adding more unlabeled videos help with the results.  It's like if I motivate a research work by stating that labeling images is hard for the task that we have but then show that imagenet without labels is quite enough for our task. Then a natural question is why did we ignored the labels in the first place?  Overall, I find this as a motivation to be tacked on which is mostly trying to "justify" the unused labels rather than raising a valid point on why.

---

> > > > ### Author Response · Authors · 2022-11-22
> > > > **Authors response to hyJV #2**
> > > >
> > > > We are glad the updates addressed some of your concerns. Thank you for increasing the rating based on the additional details and experiments, and for taking the time to engage in a constructive conversation.
> > > >
> > > > On evaluation: Got it. Thank you for recognizing that we did our best, and we agree that more work needs to be done in the community to improve evaluation protocols for generative models.
> > > >
> > > > Failure cases: Agreed, thanks for the feedback. We will add examples of the failure cases discussed in the paper to the website.
> > > >
> > > > Unlabeled videos: Understood. While 10M unlabeled videos resulted in significantly better video generation results than we anticipated, (short) video generation is not a solved problem! There is a good chance using more videos will improve performance. But if an approach relies on captioned videos, it will not be feasible to scale. While it would indeed be better if we already had the scaling result, (most works have a future work that would be ideal if it was already done? :)), this is indeed part of our future work.
> > > >
> > > > On CogVideo: Note that we perform evaluation with both English and Chinese (translated from English) prompts. We have actually been in conversation with some of the CogVideo authors through this work. It was per their suggestion that we set up the English evaluation the way we did. Based on automatic metrics on MSR-VTT and UCF zero-shot, we found English prompts to perform better. In human evaluation, the results were more mixed. English prompts had better faithfulness while Chinese prompts had slightly better quality. Considering the margins, English prompts were overall better. That is why we showed videos generated using English prompts on the website.
> > > >
> > > > Not directly relevant to your concerns about non-Chinese evaluation, but in case it is helpful, some additional notes on evaluation with Chinese prompts: A native speaker sanity checked some of the English → Chinese translations used in the human evaluation. Also, for the 28 videos shown on the website, the prompts are from Video Diffusion Models’ website, and the Chinese translations for them were all sanity checked by a native speaker (although as discussed above, we finally showed results using English prompts on the website). Moreover, for UCF zero-shot evaluation, the prompts are fairly simple, so the translations were high quality (and yet, Make-A-Video significantly outperforms CogVideo). Some of the DrawBench prompts are intended to check a model’s robustness to typos. This hurts the translation significantly, and so we manually corrected those prompts before translating them so as to not disadvantage CogVideo.
> > > >
> > > > Note that the metrics reported on UCF-101 finetuning setting are directly from the CogVideo paper. So there was no intervention from us in that evaluation.
> > > >
> > > > All in all, we agree that this setup is not ideal. Related to the above point on evaluation, we tried our best to do due diligence and sanity checks to make sure the evaluation is as meaningful as possible given the current benchmarking protocols in existing literature.

---

### Official Review · Reviewer_SMdY · 2022-11-02

**Confidence:** 4
**Clarity, Quality, Novelty And Reproducibility:** 1. The main contribution of this pape…
**Correctness:** 3
**Technical Novelty And Significance:** 3
**Empirical Novelty And Significance:** 2
**Recommendation:** 8

**Strength And Weaknesses:**

Strength:
1. Novel idea on training text-image and image-video separately to bypass paired text-video data;
2. Well-motivated to use text-image model for text-video generation;
3. State-of-the-art results;

Weaknesses:
1. The main idea is novel to combine text-image generation and image-video reconstruction; however, text-image pipeline is exactly same as previous work, and it's hard to say the components for image-video generation are novel as they are commonly used.
2. Results and analysis are not sufficient. The results are state-of-the-art but it's not clear where the improvements come from. At least some quantitative results on ablation internally should be provided to make the model less blackbox. For example, how much improvement does FRAME INTERPOLATION NETWORK bring in terms of the evaluation on MSR-VTT and UCF-101, or more qualitative examples to show the difference in the generated videos w/ and w/o the module. Otherwise it's really hard to understand how and why the model works.
3. Some claims are unclear and not well-supported: In Previous Work, "Second, we fine-tune the T2I model for video generation, gaining the advantage of adapting the model weights effectively, compared to freezing the weights as in CogVideo". This is not completely true: first, only some components of T2I model are tuned for image-video generation, starting from Dt; second, some more results should be provided to prove that in Make-A-video, fine-tuning the weights in T2I is better than freezing them, other than comparing with CogVideo.


**Summary Of The Paper:**

This paper introduces Make-A-Video, a text-video generation approach trained based on a text-image model. The core idea of Make-A-Video is to take use of learned text-vision correlation from well-trained text-image model to accelerate the learning of text-video generation. It also claims that no paired text-video data is required for video generation. The quantitative results on MSR-VTT and UCF-101 are state-of-the-art and better than CogVideo and VDM, which are two main baselines to compare. More qualitative examples are provided to elaborate the motion consistency and richer content in the generated videos from Make-A-video.

**Summary Of The Review:**

Overall, the paper introduces an idea to break the dependency on text-video pairs for text-video generation and achieves state-of-the-art performance. However, some training and inference details are missing; more analysis on the components of the model internally should be provided to elaborate why the model works, as it seems all quantitative and qualitative results are compared with other approaches but internal analysis is missing.

---

> ### Author Response · Authors · 2022-11-14
> **Authors response to reviewer SMdY**
>
>
> > “At least some quantitative results on ablation internally should be provided to make the model less blackbox. For example, how much improvement does FRAME INTERPOLATION NETWORK bring in terms of the evaluation on MSR-VTT and UCF-101, or more qualitative examples to show the difference in the generated videos w/ and w/o the module. Otherwise it's really hard to understand how and why the model works.”
>
> Thank you for the suggestion. We have included an ablation study in the updated revision (appendix).  It is not ideal to conduct these evaluations on MSR-VTT and UCF-101 because MSR-VTT metrics (CLIPSIM and FID) are frame-level metrics that do not consider temporal information. UCF-101 is too constrained and the evaluation of FVD is limited to 16 frames at 224 resolution. Therefore, we conduct human evaluation to validate the effectiveness of the different components. As shown in Table 6, our interpolation model and two SR models are all very important to improve the video quality. Specifically, 92.48% raters prefer videos of 16x256x256 over 16x64x64; 68.30% raters prefer videos of 76x256x256 over 16x256x256; 60.13% raters prefer videos of 16x768x768 over 16x256x256.
>
> > “some more results should be provided to prove that in Make-A-video, fine-tuning the weights in T2I is better than freezing them, other than comparing with CogVideo.”
>
> According to CogVideo authors, the original intent to freeze the weights was that“.. the temporal attention follows a different attention pattern and quickly ruins the pretrained weights during the initial phase of training with large gradients.”. It seems that our model is better conditioned and is able to train all weights of the attention modules jointly. Hence, freezing part of the model was not found to be necessary.
>
>
> **Clarifications:**
>
> > The main contribution of this paper seems from image-video generation, where the input images are sampled from the videos, as in 3.4; no contribution and novel components in text-image part. Please clarify this.
>
> Correct. We’ve indicated in Section 3.1 that we are “sharing the core components with the work of (Ramesh et al., 2022)”.
>
> > In 3.2, "In qualitative inspection we found this to significantly outperform per-frame super resolution", more qualitative examples should be provided for this claim.
>
> In Section 6.3, we provide an ablation study that shows that temporal super-resolution outperforms per-frame super-resolution, according to human preferences.
>
> > It seems SRh is completely same as in (Ramesh et al., 2022). Make-A-video only operates on multiple frames separately to increase the resolution. If so, the contribution as in the claim "high resolution" is not really from this work but from the text-image model in (Ramesh et al., 2022). Please clarify.
>
> $SR_l$ is spatio-temporal, whereas $SR_h$ is only spatial, as in (Ramesh et al., 2022).
>
> > Some training details are missing: the objective to train video generation; also for text-image generation. It's true that many works use them commonly but it's worth mentioning e.g., which objective function is used for the training of the components.
>
> Please see equation 4 in the updated revision.

---

### Official Review · Reviewer_X9uC · 2022-11-03

**Confidence:** 4
**Correctness:** 3
**Technical Novelty And Significance:** 2
**Empirical Novelty And Significance:** 2
**Recommendation:** 5

**Clarity, Quality, Novelty And Reproducibility:**

This paper is well written and easy to follow. The proposed method is reasonable. Considering the simplicity of the method, the reproducibility should be high.

**Strength And Weaknesses:**

Strength:
1. The paper is well written and easy to follow.
2. The proposed method shows better quantitatively results, compared to the baselines, and also have a good human evaluation.
3. The training of proposed method does not need text-video pairs, which can avoid the difficulty to have a text-video dataset.

Weaknesses:
1. Basically, the proposed method relies on many previous works: (1) the T2I model used in the proposed method is DALL-2, (2) the proposed Pseudo-3D convolutional layers is based on separable convolutions (Chollet, 2017), actually a similar idea to separate spatial and channel information to reduce computational cost have been widely adopted in different areas, and (3) as mentioned by authors, factorized space-time attention layers have also been used in VDM (Ho et al., 2022) and CogVideo (Hong et al., 2022).
2. Compared to CogVideo, the better performance achieved by the proposed method might benefit from a powerful pretrained T2I model (i.e., DALLE-2). If authors use CogView as the T2I model, same as CogVideo, could the proposed method still have a better performance?
3. I am confused about the quality of synthetic video results, as authors only show a small number of frames for each video results, and the frames in the same video only show limited differences between each other. Also, current text-to-image generation methods, such as DALLE-2, can produce loads of images with some differences, and combine these results together, we can also observe some meaningful story from them.
4. Authors claim their method accelerates training of the T2V model, but there are no experiments to support this claim.

**Summary Of The Paper:**

The paper focuses on text-to-video generation, but without having text-video pairs. To achieve this, the proposed method relies on a pre-trained text-to-image generation model, and then further extend it to generate a video, semantically aligned with the given text.

**Summary Of The Review:**

Although the proposed can produce a good results without using text-video pairs, it relies on many previous works and has limited differences between them.

---

> ### Author Response · Authors · 2022-11-14
> **Authors response to reviewer X9uC**
>
>
> >Basically, the proposed method relies on many previous works .. a similar idea to separate spatial and channel information to reduce computational cost have been widely adopted in different areas
>
> Indeed there has been substantial work to separate spatial and temporal information, and we did our best to include those in the previous work section. The additional 3x1x1 convolution projection (after each 1x3x3) such that the temporal information will also be passed through each of the U-Net’s convolution layers, is a marginal contribution that empirically was found useful for training the system. Our main contribution is to extend a T2I framework and condition on the first frame embedding to generate a video, and thus bypass the need for text-video data, which has not been demonstrated before. The proposed model architecture has added temporal modeling capabilities. We have also added an additional ablation study in Section 6.
>
> >If authors use CogView as the T2I model, same as CogVideo, could the proposed method still have a better performance?
>
> We thank the reviewer for this suggestion. We have conducted a study to disentangle the contributions of the T2I and temporal components in Make-A-Video. We did this by replacing components with one of the leading baselines, CogVideo. As can be seen in Section 6.1 of the updated revision, our method is favored across all metrics when compared to two of its CogVideo variants. Human evaluators preferred generations that used Make-A-Video's spatial-temporal mechanism (I2V) over CogVideo's about 2 out of 3 times. Similarly, evaluators preferred generations using Make-A-Video's T2I module over CogVideo's 3 out of 4 times. Please see the full results in Table 4 in Section 6.1 of the updated revision.
>
> >I am confused about the quality of synthetic video results .. current text-to-image generation methods, such as DALLE-2, can produce loads of images with some differences
>
> A video is not just a series of images that are all similar with slight differences -- we are after generating semantically meaningful motions that are temporally coherent.
>
> The supplementary web page includes 157 videos of selected samples & visual comparisons to previous works. The authors believe that adding more samples per query would clutter the page. We will consider alternative ways to add more samples. Also, as hyJV mentions, it is always hard to show videos in PDF format, whereas a web page (https://gen-videos.github.io/) is more effective where a video can be viewed as a video.
>
> >Authors claim their method accelerates training of the T2V model, but there are no experiments to support this claim.
>
> In the updated revision (Section 6.2), we compare the performance of our T2V model when trained from scratch (T2I weights are randomly initialized) on WebVid and observe a x2 slow down compared to a T2V model initialized with a pre-trained T2I. Specifically it takes 100K iterations for the randomly-initialized model to reach a CLIPSIM score of 0.246, whereas our model achieves the same score at 50K iterations.

---

### Author Response · Authors · 2022-11-14
**We thank all the reviewers for the helpful feedback**

X9uC, hyJV & 5k2D agree the paper is well written, easy to follow, and the idea behind it is well-motivated and interesting.  SMdY agrees the proposed method is novel & shows state of the art results. 5k2D agrees we present a new method for text to video generation, which is a really important problem.

X9uC & hyJV have both advised to add an additional ablation study with respect to the CogVideo baseline. We hope the disentangling efficacy study we updated in Section 6 sheds more light on this.

More broadly, we generally agree the field “lacks standard benchmark” as both hyJV & 5k2D pointed out. That said, we believe our empirical evaluation goes far beyond what is done in most existing works, and includes not only a comparison with existing commonly used metrics, but also extensive human evaluation, including a comparison to examples these works shared (or generated using models they released). We also included a webpage of many generated examples. We hope the community will continue making progress towards better benchmarks for generative models. We hope the test prompts we plan to release (as mentioned in Section 1) is a step in that direction.

---

### Author Response · Authors · 2022-11-22
**We would like to thank the reviewers again for taking the time to review our paper.**

Thank you hyJV for the additional thoughts! As we are approaching the end of the discussion period, please let us know if any of the other reviewers have additional thoughts or concerns. We would be happy to address them.

---

### Decision · Program_Chairs · 2023-01-20

**Decision:**

Accept: poster

**Justification For Why Not Higher Score:**

It's a good poster.

**Justification For Why Not Lower Score:**

Two out of four reviewers have raised their recommendation after the rebuttal to 6 and 8. The concerns of one reviewer that put 5 have been addressed in the rebuttal. AC agrees with the majority of the reviewers and recommends acceptance.

**Metareview: Summary, Strengths And Weaknesses:**

This paper proposes a text-to-video generation framework. It builds upon advances in text-to-image synthesis and brings the generated images alive by learning motion from a large corpora of unlabeled video data. The proposed architecture with a spatiotemporally factorized diffusion model is evaluated extensively using automatic metrics and human evaluation.
A strong rebuttal has cleared out several critical issues raised by the reviewers; hence two reviewers have raised their scores. Also the rebuttal has addressed the concerns raised in the review with a below borderline score. AC recommends acceptance of the paper, and urges the authors to polish their final version with any remaining questions. Congratulations to the authors!

**Note From Pc:**

if the above contains the word "oral" or "spotlight" please see: "oral" presentation means -> notable-top-5% and "spotlight" means -> notable-top-25%. As stated in our emails, we are disassociating presentation type from AC recommendations